# Construction 7-membered ring via Ni–Al bimetal-enabled C–H cyclization for synthesis of tricyclic imidazoles

Jiang-Fei Li[1], Wei-Wei Xu[1], Rong-Hua Wang[1], Yue Li[1], Ge Yin[1] & Mengchun Ye [1✉]

The construction of 7-membered ring via direct C7–H cyclization of benzoimidazoles with alkenes would provide a more atom- and step-economical route to tricyclic imidazoles and derivatives that widely exist in a broad range of bioactive molecules. However, transition metal-catalyzed C–H cyclization for medium-ring synthesis has been limited to reactive C–H bonds, instead, the activation of unreactive C–H bonds towards medium synthesis still remains an elusive challenge. Herein, we report a direct construction of 7-membered rings via Ni–Al co-catalyzed unreactive C7–H cyclization of benzoimidazoles with alkenes, providing a series of tricyclic imidazoles in 40–98% yield and with up to 95:5 er.

[1] State Key Laboratory and Institute of Elemento-Organic Chemistry, College of Chemistry, Nankai University, Tianjin, China. ✉email: mcye@nankai.edu.cn

Tricyclic imidazoles and derivatives bearing a 7-membered ring are an important class of structural motifs that widely exit in diverse range of bioactive and material molecules (Fig. 1a)[1–7]. However, due to the difficulty in construction of medium rings that requires overcoming unfavorable entropy and transannular strain[8–10], synthetic routes of such tricyclic imidazoles are quite limited. A typical method relies on Friedel-Crafts

acylation for cyclization, often requiring multiple synthetic steps and stoichiometric amounts of AlCl₃ catalyst (Fig. 1b)[11–14]. Another alternative is to use alkene metathesis to form 7-membered rings, generally needing lengthy routes for both starting material preparation and final product formation (Fig. 1c)[15]. Thereby, direct construction of 7-membered ring via C7–H cyclization of benzoimidazoles with alkenes would provide

**Fig. 1 Synthesis of tricyclic imidazoles and derivatives bearing a 7-membered ring. a** Tricyclic imidazoles bearing a 7-membered ring in bioactive molecules and materials. **b** Typical method I: Friedel-Crafts reaction. **c** Typical method II: alkene metathesis reaction. **d** Medium ring synthesis via transition metal-catalyzed cyclization of reactive C–H bonds. **e** Ni–Al bimetal-catalyzed direct C7–H cyclization for synthesis of tricyclic imidazoles bearing a 7-membered ring (this work).

**Fig. 2 Reaction optimization.** Reaction conditions: **1a** (0.20 mmol), toluene (1.0 mL), under N₂ for 3 h. Yield was determined by ¹H NMR analysis with CH₂Br₂ as the internal standard. IPr = 1,3-bis(2,6-diisopropylphenyl)-2,3-dihydro-1*H*-imidazole. Cy₃P = triisopropylphosphine. dppe = 1,2-bis (diphenylphosphino)ethane. BINAP = 2,2'-bis(diphenylphosphino)-1,1'-binaphthalene. IMes = 1,3-dimesityl-2,3-dihydro-1*H*-imidazole. SIPr = 1,3-bis(2,6-diisopropylphenyl)imidazolidine. DME = 1,2-dimethoxyethane.

a more straightforward, atom- and step-economical access to tricyclic imidazoles from more easily-accessible substrates. However, transition metal-catalyzed C–H cyclization for medium-ring synthesis has been a challenging goal during the past two decades[16,17]. Early efforts focused on the cyclization of reactive formyl C–H bonds with alkenes via rhodium catalysis[18–28]. Until recent years, non-formyl C–H bonds were also able to be activated via nickel catalysis to form 7-membered rings (Fig. 1d)[29–34], while these examples were still limited to reactive C–H bonds such as heterocyclic and polyfluoro-aromatic C–H bonds, and moreover, only scattered substrates were reported with in general low to moderate yield and ee. In contrast, the activation of prevalent and unreactive aromatic C–H bonds towards 7-membered ring synthesis still remains an elusive challenge. The difficulty was ascribed to the fact that unreactive aromatic C–H bonds are often more reluctant to be activated by low-valent metals owing to their higher bond strength and weaker acidity[35].

Here, we show that the construction of 7-membered ring via Ni–Al bimetal-catalyzed unreactive C7–H bond cyclization of benzoimidazoles with alkenes is achieved, providing a series of tricyclic imidazoles in 40–98% yield and with exclusively *endo* selectivity and up to 95:5 er (Fig. 1e). In this reaction, the use of Ni–Al bimetallic synergistic catalysis instead of traditional mono-metal catalysis[36–47] greatly facilitated the formation of medium rings owing to the following two reasons: (1) the coordination of Al-Lewis acid to the N atom of imidazoles would contribute to decreasing electron density of the aromatic ring, thus promoting C–H bond activation; (2) proper steric hindrance from

C2 substituent of benzoimidazoles would reduce unfavorable entropy effect, favoring the formation of medium rings.

## Results

**Reaction optimization.** We commenced our study by selecting benzoimidazole **1a** as a model substrate, nickel as a catalyst and Al-Lewis acid as a co-catalyst (Fig. 2). A systematic survey on Ni metals, Al Lewis acids, ligands, bases, and other reaction parameters led to the optimal conditions: 10 mol% of Ni(cod)₂, 10 mol% of IPr·HCl, 10 mol% of AlMe₃ and 40 mol% of *t*BuOK in toluene at 130 °C, under which an *endo* cyclization was exclusively achieved, providing tricyclic imidazole **2a** bearing a 7-membered ring in 98% yield (entry 1).

Control experiments showed that the combination of Ni, IPr, AlMe₃ and *t*BuOK is critical, and the removal of any of them would greatly reduce the yield (entries 2–5). Traditional phosphine ligands such as monophosphines and bidentate phosphines were all ineffective (entries 6 and 7), whereas other N-heterocyclic carbenes were still compatible, albeit with a little lower yields (entries 8 and 9). In addition, in situ formed Ni(0) was also an effective catalyst, yet providing only 42% yield (entry 10). Base acted as another critical role in the reaction. *t*BuOLi was inefficient, whereas *t*BuONa worked well, affording a comparable result to that of *t*BuOK (entries 11 and 12). Notably, more than 10 mol% of *t*BuOK was essential to the reactivity (entries 13–15). The use of 10 mol% of *t*BuOK gave no products (entry 13), instead, leading to an imidazole with free NH group in 5% yield, which was formed from the decomposition of alkene-

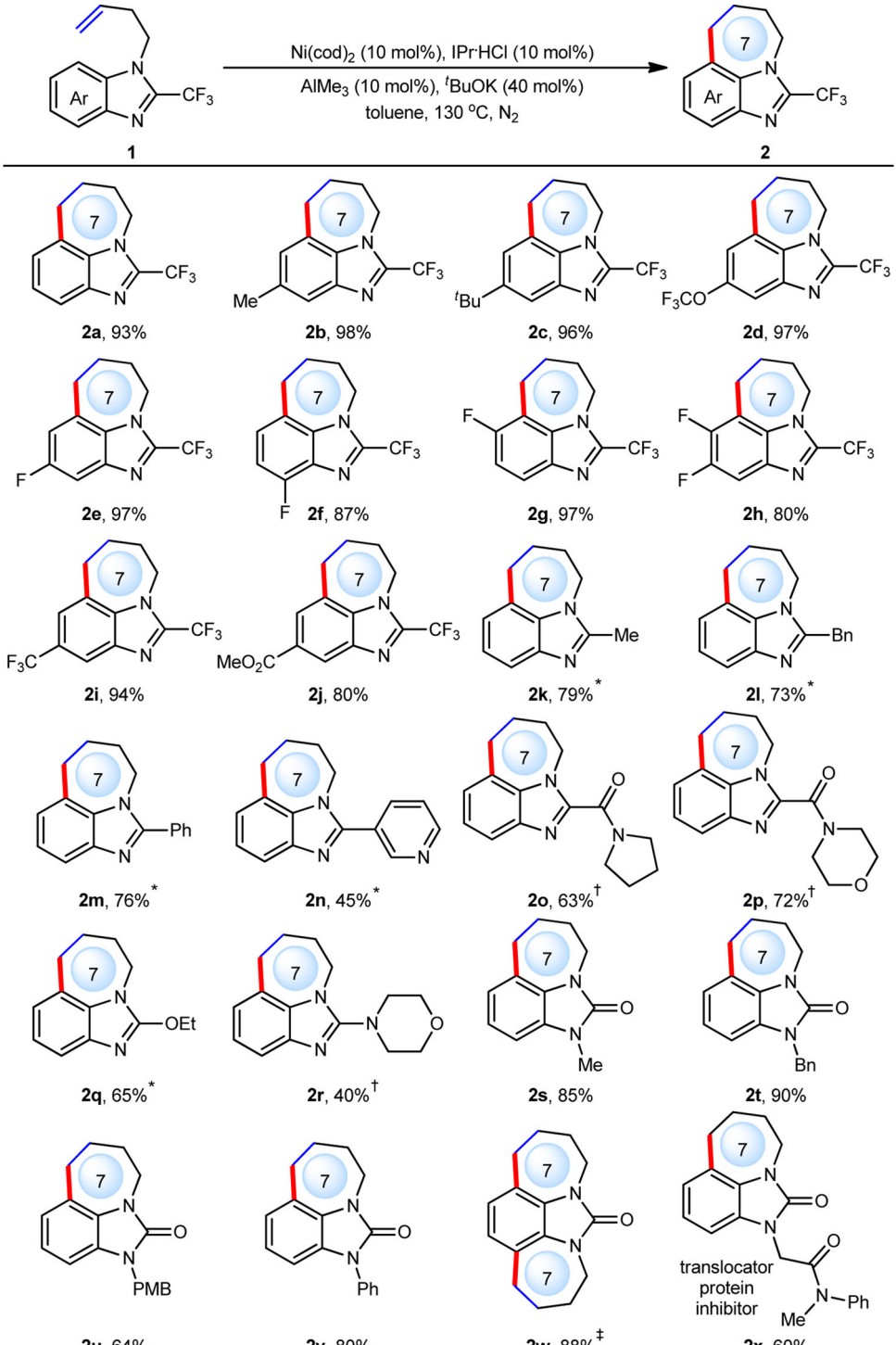

**Fig. 3 Scope of imidazoles.** Reaction conditions: **1** (0.20 mmol), toluene (1.0 mL) under $N_2$ for 3 h. Yield of isolated products. *AlMe$_3$ (200 mol%) was used. †AlMe$_3$ (60 mol%) was used. ‡AlMe$_3$ (100 mol%) and ${}^t$BuOK (80 mol%) were used.

isomerization substrate. We reasoned that excess ${}^t$BuOK could suppress the isomerization of the terminal alkene as the literature proposed[48].

**Scope of imidazoles and alkenes.** With the optimized conditions in hand, various benzoimidazole motifs bearing different substituents on the aromatic ring were investigated first (Fig. 3). Results showed that either electron-donating groups such as methyl (**2b**) and *tert*-butyl (**2c**) or electron-withdrawing groups

such as CF$_3$O (**2d**), F (**2e** to **2h**), CF$_3$ (**2i**) and carboxylate (**2j**) were well compatible with the reaction, providing the corresponding products in 80–98% yield. Notably, C2 substituents of benzoimidazoles proved critical to the reactivity. Without C2 substituents, C2–H cyclization would dominate to form a 6-membered ring as we previously reported[41], further suggesting that C7–H bond was quite unreactive towards Ni catalysis. In general, electron-deficient CF$_3$ group on C2 position can ensure high reactivity with using only 10 mol% of AlMe$_3$ co-catalyst,

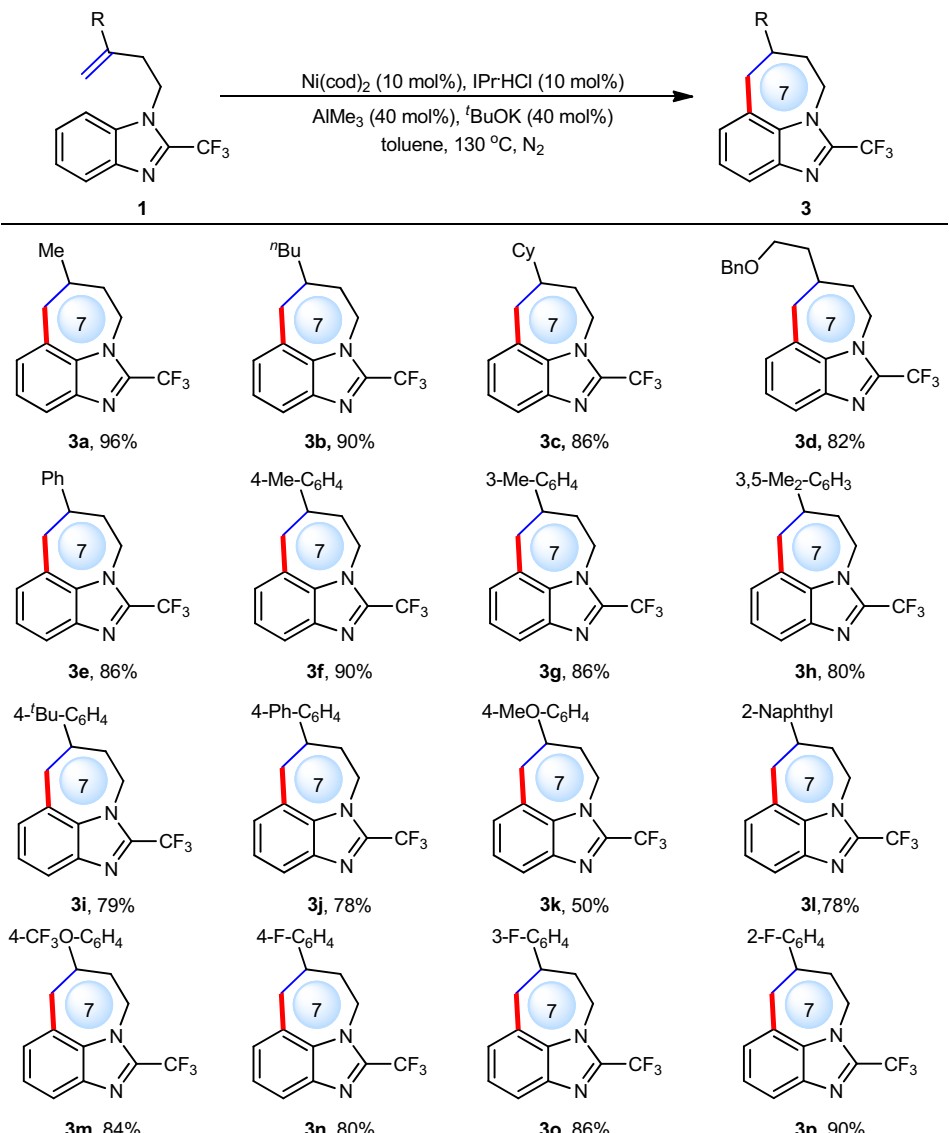

**Fig. 4 Scope of alkenes.** Reaction conditions: **1** (0.2 mmol), toluene (1.0 mL) under $N_2$ for 3 h. Yield of isolated products.

while $CF_3$ group was not indispensable, and it can be replaced by a broad range of other substituents such as alkyl (**2k** and **2l**), (hetero)aryl (**2m** and **2n**), carbamoyl (**2o** and **2p**), alkoxy (**2q**) and amino (**2r**) groups, providing the corresponding products in 40–79% yield by tuning the amount of $AlMe_3$. Pleasingly, imidazole-2-ones, which also widely exist in numerous bioactive compounds, were well compatible with the current reaction.

When N-protecting groups varied from Me (**2s**), Bn (**2t**), PMP (**2u**) to Ph (**2v**), the corresponding products can be smoothly obtained in 64–90% yield. In consideration of two symmetrical N atoms in the molecule, dual C–H annulation was then investigated and a tetracyclic product (**2w**) bearing two 7-membered rings can be smoothly achieved, which is not easily accessed by traditional Friedel-Crafts reaction because the second acylation would be quite difficult. Besides simple aryl and alkyl groups, carboxylate group was also tolerated, providing a translocator protein inhibitor (**2x**) in 60% yield[1].

Next, the compatibility of alkene motifs were investigated (Fig. 4). Although internal and trisubstituted alkenes were ineffective because of big steric hindrance, various 1,1-disubstituted terminal alkenes proved to be effective. Different types of

alkyls such as methyl (**3a**), linear *n*-butyl (**3b**), branched cyclohexyl (**3c**) and functionalized alkyl (**3d**) were well tolerated, delivering the corresponding products in 82–96% yield. Considering that the incorporation of aryl motifs can significantly increase the complexity of molecules, we examined various aryl substituted alkenes (**3e–3p**). Results showed that these aryl alkenes bearing either electron-rich groups such as methyl (**3f–3h**), *t*Bu (**3i**), Ph (**3j**), methoxy (**3k**), and naphthyl (**3l**) or electron-deficient groups such as $CF_3O$ (**3m**) and F (**3n–3p**) at different positions of the aryl ring all proceeded smoothly, providing the corresponding products in 50–90% yield.

**Enantioselective attempts**. For the synthesis of medium ring, a flexible large ring transition state would be involved, rendering the enantioselective control of such a reaction quite challenging[29–34]. By surveying a wide range of chiral carbenes, we found that bulky ANIPE, previously developed by Shi and Cramer groups[49–51], was the optimal ligand (see the Supplementary Information for details). With this ligand, a series of substrates with various alkene motifs were then tested (Fig. 5). In general,

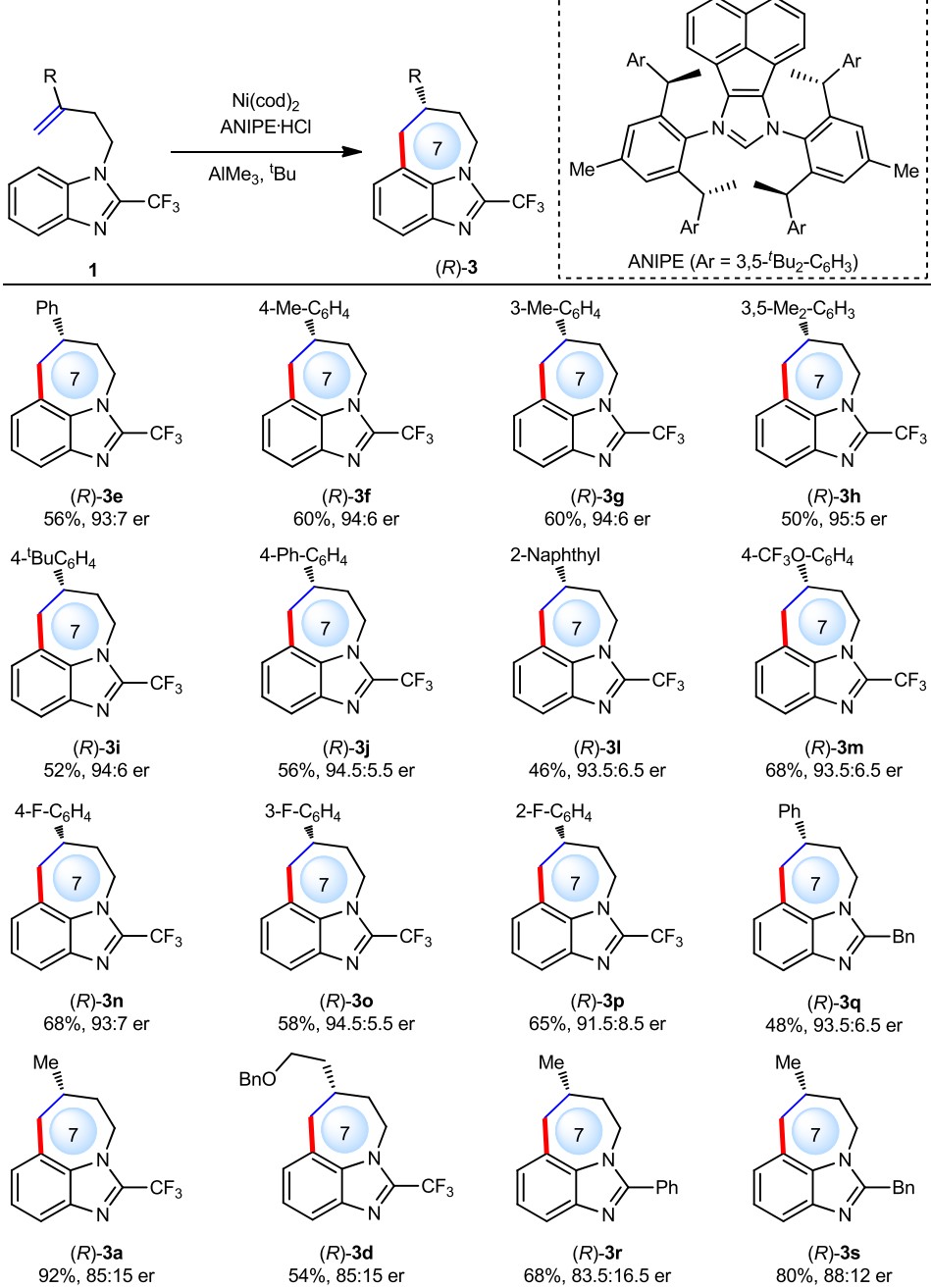

**Fig. 5 Enantioselective control.** Reaction conditions: **1** (0.20 mmol), Ni(cod)$_2$ (10 mol%), ANIPE·HCl (10 mol%), AlMe$_3$ (80 mol%), $^t$BuOK (40 mol%), toluene (1.0 mL) at 130 °C under N$_2$ for 3 h. Yield of isolated products. Ee was determined by chiral HPLC.

various aryl groups were well compatible with the current reaction, providing the corresponding products in good yields and with 91.5:8.5 to 95:5 er ((*R*)-**3e** to (*R*)-**3q**). However, alkyl groups, albeit still with good yields, would result in slightly decreased ee ((*R*)-**3a**, (*R*)-**3d**, (*R*)-**3r** and (*R*)-**3s**) owing to bigger structural flexibility. The (*R*) absolute configuration of major enantiomer of the product was determined by single crystal X-ray diffraction.

**Reaction utility and mechanistic discussion**. To demonstrate the utility of the current method, a gram-scale reaction of **1a** was conducted, and a comparable yield was obtained under the standard conditions (Fig. 6a). Tricyclic imidazole derivative **2s**

can be easily oxidized at the benzylic position to produce an intermediate **4** in 62% yield, which can be further transformed into various bioactive molecules such as β-2-adrenergic agonists and zilpaterol (Fig. 6b)[11–14]. In addition, bioactive molecule, translocator protein inhibitor (**2x** in Fig. 3), can be easily accessed from readily available imidazole-2-one through the current method.

To gain more insights into the reaction, relevant mechanistic experiments were conducted. Deuterium-labeling experiment showed that C7-D on the aromatic ring was completely transferred to the 7-membered ring, and moreover, no deuterium scrambling was observed at other positions (Fig. 6c), which suggested that an *endo*-insertion of alkene to Ni–H bond could

**Fig. 6 Synthetic utility and mechanistic experiments. a** Gram-scale reaction. **b** Product transformation. **c** Deuterium labeling experiments. **d** Kinetic isotope effect. **e** Proposed mechanism.

proceed via an irreversible step. Both competitive experiment between equivalent moles of **1a** and $d_4$-**1a** and parallel reactions revealed significant kinetic isotope effect ($k_H/k_D = 5.75$, 4.81, respectively), indicating that the C–H cleavage could be the rate-determining step (Fig. 6d), and it could proceed via oxidative addition mechanism because direct H transfer pathway in general gives low kinetic isotope effect[35]. In addition, $^{19}F$ NMR spectra of

stoichiometric reactions suggested that nickel could rapidly coordinate to the alkene motif of substrate **1a**, and then initiate next C–H cleavage and alkene insertion (see the Supplementary Information). On the basis of these facts, a plausible mechanism was proposed as below (Fig. 6e): substrate **1a** coordinates with AlMe₃ and nickel first, and then facilitates Ni-catalyzed C7–H bond cleavage via oxidative addition process. Subsequent

irreversible *endo*-type alkene migratory insertion and reductive elimination generates the Al-coordinated product, which exchanges with another substrate **1a** to initiate a next cycle.

## Methods

**General procedure for Ni-catalyzed C7–H cyclization**. To a 15 mL oven dried tube in glove box were added Ni(cod)$_2$ (5.5 mg, 10 mol%), IPr HCl (8.6 mg, 10 mol%), $^t$BuOK (9 mg, 0.08 mmol, 40 mol%), benzoimidazole **1** (0.2 mmol), dry degassed toluene (2.0 mL), and AlMe$_3$ (1.0 M/hexane, 10 mol% or 60 mol% or 200 mol%). The tube was capped, taken outside the glove box, and stirred at 130 °C for 3 h. After that, the mixture was cooled to r.t., quenched with 2 mL of 5% EDTA disodium salt solution, and filtered through a short plug of silica gel, eluting with EtOAc. The filtration was concentrated in vacuo to afford the crude product, which was further purified by flash column chromatography on silica gel (EtOAc/hexanes).

## Data availability

The authors declare that the data supporting the findings of this study are available within the article and its Supplementary Information file. For the experimental procedures, data of NMR and HPLC analysis, see Supplementary Methods in Supplementary Information file. The X-ray crystallographic coordinates for structures reported in this study have been deposited at the Cambridge Crystallographic Data Centre (CCDC), under deposition number CCDC 2009572. These data can be obtained free of charge from The Cambridge Crystallographic Data Centre via https://www.ccdc.cam.ac.uk/structures/.

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

## Acknowledgements

We thank the National Natural Science Foundation of China (21871145 and 91856104), the Tianjin Applied Basic Research Project and Cutting-Edge Technology Research Plan (19JCZDJC37900) for financial support for financial support.

## Author contributions

J.-F.L. discovered and developed the reactions. W.-W. X., R.-H. W., Y.L., G.Y. performed part of synthetic experiments. M.Y. conceived, designed the investigations and wrote the manuscript. J.-F.L. wrote the Supplementary Information.

## Competing interests

The authors declare no competing interests.
