## [Peer Review File · Nature Communications]

REVIEWER COMMENTS

Reviewer #1 (Remarks to the Author):

The paper submitted by Jiang-Fei Li, Wei-Wei Xu, Rong-Hua Wang Yue Li, Ge Yin, and Mengelhun Ye describes N-(3-buten-1-yl)benzimidazoles and its 2-substituted derivatives undergo endo-cyclization forming a 7-membered ring with the aid of Ni(cod)₂, AlMe₃ and NHC ligand. Using appropriate optically active NHC ligand, asymmetric synthesis of the resulting tricyclic heterocycles is achieved with good enantioselectivity. Because such heterocyclic framework is found in biologically active agents, the importance of the net transformation is emphasized in view of simple experimental operation and step economy.

The total transformation is understood in terms of coordination of Al to benzimidazole nitrogen followed by C7-H activation by Ni(0)-coordinated by the terminal C=C bond followed by endo-cyclic hydronickelation to the C=C bond and oxidative coupling. Since the mechanistic picture proceeding through C-H activation of heterocycles by Ni-Al dual catalysis followed by coupling to terminal alkenes is well-described by Nakao and Hiyama (refs. 36, 37) and Nakao and Hartwig (ref. 45), it is not surprising to anticipate the present reaction might work. This paper is definitely a good addition to the chemistry of Ni-Al dual catalysis.

Accordingly, the paper looks better suit with a journal that deals with organic synthesis.

Reviewer #2 (Remarks to the Author):

Ye and co-workers describe a Ni/Al co-catalyzed intramolecular C-H alkylation of benzoimidazoles with alkenes in this publication. A series of tricyclic imidazoles and derivatives bearing a 7-membered ring were synthesized rapidly and straightforwardly. The combination of nickel/NHC with Lewis acidic aluminum co-catalyst is the key to promote the reaction effectively. The yields and functional group tolerance are good; the substrate scope is considerably broad. Moreover, the authors report an enantioselective variant of the transformation using a bulky chiral ANIPE-type NHC ligand previously developed by Shi and Cramer groups. Good to high levels of enantioselectivity is achieved. They nicely conducted mechanistic studies to propose a possible catalytic cycle containing Ni-coordination, the rate-determining C-H cleavage, alkene hydronickelation, and reductive elimination. This work represents a rare and efficient enantioselective C-H functionalization method for the construction of heterocycles bearing 7-membered rings. I recommend the acceptance of this well-planned and better-executed transformation (the SI is certainly exquisite) in Nature Communications.

Minor comments:

1. The abbreviations of chiral NHCs, for example, "AnIPe" in the main text and "AnIPr-1" in the SI need to be changed to "ANIPE" and "ANIPE-1", respectively.
2. When the authors comment about the ANIPE-type ligand, credits are encouraged to give to Shi and Cramer by writing, for example, "the ANIPE-type ligand previously developed by Shi and Cramer groups".
3. What if trisubstituted alkenes are employed as substrates? Readers of the journal would appreciate that the authors make a comment on the utilization of trisubstituted olefin. Probably a simple comment on the reference section might suffice.
4. For better understanding the challenging and exciting enantiocontrol in the formation of 7-membered rings, stereochemical models are welcomed to include in either the main text or SI, if possible.

Reviewer #3 (Remarks to the Author):

This manuscript by Ye and coworkers described the preparation of 7-membered ring via direct intramolecular C7-H cyclization of benzimidazoles with its dangling alkene using catalytic amount

of Ni(COD)₂, NHC carbene, Al Lewis acid and tBuOK base. A synthetic method relied on simple C-H bond activation to make potential bioactive tricyclic imidazole is the important step toward the realization of atomic-economy strategy. Thus, the work based on bimetallic mediated selective C-H bond activation would become common tool of utility in the future organic synthetic arsenal, despite the main concept for remote C-H activation by Ni/Al developed for more than 10 years. Variety of substrates (40 examples) in term of imidazole and alkenes derivatives are suitable in this catalytic protocol. Moreover, a high enantioselectivity was also witnessed by using chiral NHC ligand. Authors also included the results of mechanistic investigations, which was consistent with the proposed mechanism. This reviewer supported the publication of this work in this high-profile journal Nature Communication only if the following issues have been properly addressed in satisfactory manner.

1. Why this Ni-catalyzed C-H activation reaction went anti-Markovnikov hydrogen addition on terminal alkene to construct 7-membered ring, not the formation of 6-membered ring?
2. In this reaction, AlMe₃ as Lewis acid and electron-withdrawing group such as CF₃ is able to weaken the C7-H bond for activation. What is the reason that 2 equiv. of AlMe₃ is used in the electron-donating substituted cases (Figure 4 2k-m)? 1 equiv. AlMe₃ is theoretically enough due to only one chelated site.
3. In the optimization, any other Lewis acids (for example: AlPh₃) have been tried for controlling the side-selective of imidazole?
4. In Figure 6 (enantioselective control), the label of compounds should change, it cannot share the compound label with racemic scope (Figure 5). (e.g. JACS, 2021, doi: 10.1021/jacs.1c00622)
5. Although 7-membered ring construction is the principal research in this paper, is it possible to extend to construct 6- or 8-membered ring with this methodology?
6. In supporting information: wrong page number in table of contents; double comma in 1m(Line 1, p.S4), 1n(Line 2, p.S4), 1w(Line 2, p.S5), 1m'(Line 1,p.S14); data missing in 1v (state of compound, p.S5); incorrect typing: ArB(OH)₂ (first scheme, p.S10); missing compound label in asymmetric control(p.S32-p.S36).
7. In supporting information: missing all the yields in precursor synthesizes; ¹H NMR peak labels overlapped in several cases.
8. Figure 5. Scope of alkyne should be replaced by scope of alkene?
9. Please include appropriate additional references for Ni-Al selective CH bond activation: Chem. Comm. 2015, 51, 17104–17107, Org. Lett. 2014, 16, 4826–4829.
10. Authors should include its synthetic utility into making biological active molecule or drug precursors.

Point-to-Point Response

Response to Reviewer #2: Great thanks for the helpful comments

1. *“The abbreviations of chiral NHCs, for example, “AnIPe” in the main text and “AnIPr-1” in the SI need to be changed to “ANIPE” and “ANIPE-1”, respectively.”*

Response: "AnIPe" has been changed to "ANIPE" in main text and the SI.

2. *“When the authors comment about the ANIPE-type ligand, credits are encouraged to give to Shi and Cramer by writing, for example, “the ANIPE-type ligand previously developed by Shi and Cramer groups”.”*

Response: Related sentence “bulky ANIPE, previously developed by Shi and Cramer groups” has been added into the text.

3. *“What if trisubstituted alkenes are employed as substrates? Readers of the journal would appreciate that the authors make a comment on the utilization of trisubstituted olefin. Probably a simple comment on the reference section might suffice.”*

Response: Both internal and trisubstituted alkenes were ineffective because of big steric hindrance, and the corresponding sentence has been revised into “Although internal and trisubstituted alkenes were ineffective because of big steric hindrance”.

4. *“For better understanding the challenging and exciting enantiocontrol in the formation of 7-membered rings, stereochemical models are welcomed to include in either the main text or SI, if possible.”*

Response: Stereochemical model has been added to the supplementary information (see Supplementary Note 5).

Response to Reviewer #3: Great thanks for the helpful comments

1. *“Why this Ni-catalyzed C-H activation reaction went anti-Markovnikov hydrogen addition on terminal alkene to construct 7-membered ring, not the formation of 6-membered ring?”*

Response: The anti-Markovnikov selectivity was mainly ascribed to the presence of bulky carbene ligand, which favored the formation of a less sterically-hindered 8-membered nickelacycle through hydrogen addition on internal position of the alkene. Instead, Markovnikov addition through hydrogen addition on the terminal position of the alkene would lead to a more sterically-hindered 7-membered nickelacycle bearing a secondary or tertiary alkyl nickel. The same selectivity also appeared in intermolecular C-H alkylation of simple arenes with terminal alkenes reported by Hartwig and Cramer et al. (see ref 34 and 35).

2. *“In this reaction, AlMe₃ as Lewis acid and electron-withdrawing group such as CF₃ is able to weaken the C7-H bond for activation. What is the reason that 2 equiv.*

of AlMe₃ is used in the electron-donating substituted cases (Figure 4 2k-m)? 1 equiv. AlMe₃ is theoretically enough due to only one chelated site.”

Response: For Me-substrate (**2k**), loadings of AlMe₃ have significant effect on the yield: 1.0 equiv (48% NMR yield), 1.5 equiv (61% NMR yield), 2.0 equiv (83% NMR yield) and 2.5 equiv (84% NMR yield) (see supplementary Table 7), meaning that 2 equiv of AlMe₃ were requisite for the optimal yield. We reasoned that extra 1.0 equiv of AlMe₃ could interact with the base (^tBuOK) and its acid (^tBuOH), forming Al-O complexes with varying ratio of ^tBuO and AlMe₃. These Al-O complexes cannot activate non-active substrates for C-H activation, while still being effective for active substrates, for example, model CF₃-substrate **1a**.

3. *“In the optimization, any other Lewis acids (for example: AlPh₃) have been tried for controlling the side-selective of imidazole?”*

Response: Other Lewis acids such as AlPh₃, MgⁿBu₂ and ZnMe₂ have been examined and the results have been added into Supplementary Table 4. The results showed that these Lewis acids still worked to give the same C7-alkylated product with slightly lower yields, suggesting that the steric hindrance of Lewis acids did not have influence on the site-selectivity. We reasoned that this result could be attributed to long distance between the Lewis acid and the reaction site, as well as an intramolecular reaction model, which would prevent from generating other site selectivity owing to big ring strain. However, intermolecular reaction may have different site selectivity in the presence of different sterically-hindered Lewis acids, and this research is underway in our lab.

4. *“In Figure 6 (enantioselective control), the label of compounds should change, it cannot share the compound label with racemic scope (Figure 5). (e.g. JACS, 2021, doi: 10.1021/jacs.1c00622)”*

Response: The error has been revised.

5. *“Although 7-membered ring construction is the principal research in this paper, is it possible to extend to construct 6- or 8-membered ring with this methodology?”*

Response: We have tried to make 6- or 8-membered ring by this method, but failed. In case of the formation of 6-membered ring, an allylic amine motif is incompatible with Ni(0) conditions, easily resulting in substrate decomposition. In case of the formation of 8- or 9-membered ring, no reaction was observed even at elevated temperature (160 °C), probably owing to very low reactivity. To form larger rings, more powerful carbene ligands need to be designed to enhance reactivity.

6. *“In supporting information: wrong page number in table of contents; double comma in 1m(Line 1, p.S4), 1n(Line 2, p.S4), 1w(Line 2, p.S5), 1m’(Line 1,p.S14); data missing in 1v (state of compound, p.S5); incorrect typing: ArB(OH)₂ (first scheme, p.S10); missing compound label in asymmetric control(p.S32-p.S36).*

Response: All these errors have been revised.

7. *“In supporting information: missing all the yields in precursor synthesis; ¹H NMR peak labels overlapped in several cases.”*

Response: Yields of precursors have been added.

8. *“Figure 5. Scope of alkyne should be replaced by scope of alkene?”*

Response: The error has been revised.

9. *“Please include appropriate additional references for Ni-Al selective CH bond activation: Chem. Comm. 2015, 51, 17104–17107, Org. Lett. 2014, 16, 4826–4829.”*

Response: The suggested literatures have been added as Ref 39 and 40.

10. *“Authors should include its synthetic utility into making biological active molecule or drug precursors.”*

Response: Bioactive molecule (translocator protein inhibitor **2x** in Fig. 4) has been synthesized by the current method. But owing to the limitation of figure size, we had to put this compound synthesis into Fig. 4. To highlight this utility, we have inserted a related sentence into the utility section as “In addition, bioactive molecule, translocator protein inhibitor (**2x**), can be easily accessed from readily available imidazole-2-one through the current method (see Fig. 4)”.

REVIEWERS' COMMENTS

Reviewer #2 (Remarks to the Author):

All of my concerns have been addressed in the revised manuscript.

Reviewer #3

Reviewer #3 made comments to the editor only- all concerns have been addressed.